# Sphingosine 1-Phosphate Modulation in Inflammatory Bowel Diseases: Keeping Lymphocytes Out of the Intestine

**DOI:** 10.3390/biomedicines10071735

**Published:** 2022-07-19

**Authors:** Arianna Dal Buono, Roberto Gabbiadini, Ludovico Alfarone, Virginia Solitano, Alessandro Repici, Stefania Vetrano, Antonino Spinelli, Alessandro Armuzzi

**Affiliations:** 1IBD Center, Department of Gastroenterology, IRCCS Humanitas Research Hospital, 20089 Rozzano, Milan, Italy; arianna.dalbuono@humanitas.it (A.D.B.); roberto.gabbiadini@humanitas.it (R.G.); ludovico.alfarone@humanitas.it (L.A.); virginia.solitano@humanitas.it (V.S.); 2Department of Biomedical Sciences, Humanitas University, 20072 Pieve Emanuele, Milan, Italy; alessandro.repici@hunimed.eu (A.R.); stefania.vetrano@hunimed.eu (S.V.); antonino.spinelli@hunimed.eu (A.S.); 3Endoscopy Unit, Department of Gastroenterology, IRCCS Humanitas Research Hospital, 20089 Rozzano, Milan, Italy; 4IBD Center, Laboratory of Immunology in Gastroenterology, IRCCS Humanitas Research Hospital, 20089 Rozzano, Milan, Italy; 5Colon and Rectal Surgery Division, IRCCS Humanitas Research Hospital, 20089 Rozzano, Milan, Italy

**Keywords:** sphingosine 1-phosphate, small molecules, oral therapy, ulcerative colitis, Crohn’s disease

## Abstract

Inflammatory bowel diseases (IBDs) are chronic and disabling conditions that, uncontrolled, lead to irreversible bowel damage and associated comorbidities. Despite the new era of biological therapies, IBDs remain not curative. The treatment purpose is to induce endoscopic remission, reduce the progression of the disease and improve the quality of life. Optimal and early treatment could enable the prevention of their complications. Small molecules, administrated as oral agents, have the capacity of overcoming the limitations of biologic agents (i.e., parenteral administration, rapidity of action and primary and secondary non-responsiveness). Of special interest are results from the use of oral sphingosine 1-phosphate (S1P) receptor modulators (ozanimod, etrasimod, fingolimod and laquinimod), based on S1P activities to target lymphocyte recirculation in the mucosa, acting as immunosuppressive agents. Most S1P modulators are reported to be safe and effective in the treatment of both UC and CD. High and satisfactory rates of clinical remission as well as endoscopic improvement and remission can be achieved with these molecules. Safety alarms remain rather low, although the S1P binding to two of its G protein-coupled receptors, 2 and 3 (S1PR2 and S1PR3), may be associated with cardiovascular risks. Cost-effectiveness studies and head-to-head trials are needed to better define their place in therapy. This review summarizes these emerging data published by PubMed and EMBASE databases and from ongoing clinical trials on the safety and efficacy of selectivity of S1P modulators in the treatment of IBD.

## 1. Introduction

Inflammatory bowel diseases (IBDs), including ulcerative colitis (UC) and Crohn’s disease (CD), are chronic, disabling conditions affecting the gastrointestinal tract and characterized by an abnormal immune response to intestinal microflora in genetically susceptible individuals [1,2]. For their progressive behavior, the current recommended management involves optimal and early treatment, in order to prevent complications such as corticosteroids’ need, hospitalization, surgery and disability and dysplasia/cancer [3,4,5].

Since the advent of biologic agents targeting different inflammatory pathways (i.e., anti-tumor necrosis factor alpha, anti-integrin, anti-interleukin-12/23), the management of moderate-to-severe IBDs has been revolutionized. Indeed, their use in therapy allows steroid-free remission, induces mucosal and histological healing and lowers surgeries and hospitalizations, considerably improving the quality of life of IBD patients [6,7,8,9]. Still, several limitations of these therapies remain to be overcome: their potential immunogenicity, the parenteral administration and the high rates of loss of response/disease relapse. Particularly, around one-third to half of IBD patients initiating a biologic agent experience a primary non-response, and a further half of the patients develop a secondary loss of response [10,11,12]. Costs associated with biologics’ production and reimbursements have been further limiting the use of these agents [13]. Moreover, the better sequence of the current available agents has not yet been defined [14], and the choice of the therapy relies on a combination of the IBD phenotype and/or behavior, previous biologic treatment response and/or failure, potential adverse effects related to therapy, patient comorbidities, and, finally, the sharing of any preferences and expectations with patients themselves.

In this framework of discovering and developing more effective treatments for IBDs, novel oral small molecules, simultaneously targeting multiple cytokines (i.e., Janus kinase inhibitors) or controlling the migration of inflammatory cells, such as sphingosine 1-phosphate (S1P) receptor modulators, are currently under clinical investigation. Small-molecule drugs are characterized by a low molecular weight of less than 1 kDa and can easily diffuse through cell membranes. The oral route of administration, the main pharmacokinetic features (i.e., rapidity of onset, drug-drug interactions) and the low/absent antigenicity make these drugs particularly advantageous over biologics. Table 1 elucidates the principal characteristics and development phase of S1P modulators in IBD.

Specifically, the binding S1P with its type 1 and type 5 receptors (S1PR1, S1PR5) regulates several biological functions (cell migration, proliferation, cytokine production, survival, endothelial barrier permeability and lymphocyte trafficking [15]). The intestinal inflammation is partially maintained by the circulation of leukocytes into the intestinal wall after egressing from primary and secondary lymphoid organs, which is regulated by S1P receptors [16,17].

Among the available S1P receptor modulators, fingolimod, siponimod, etrasimod and ozanimod have already been approved by regulatory authorities for the treatment of multiple sclerosis [18,19,20]. This review highlighted S1P modulation as emerging therapies’ target in the treatment of IBDs, discussing its pharmacological features and the brand-new available data on efficacy and safety.

## 2. Methods

PubMed/MEDLINE and Embase databases were searched up to May 2022 to identify relevant studies investigating the efficacy and safety of S1P receptor modulators in the treatment of IBD. The following text words and corresponding Medical Subject Heading/Entree terms were used: “sphingosine 1-phosphate,”, “ozanimod”, “etrasimod”, “amiselimod” and “fingolimod”, individually and in combination with “inflammatory bowel disease(s)”, “IBD”, “ulcerative colitis” and “Crohn’s disease”. No publication date restrictions were applied. Articles were included in this review based on their relevance; additional publications were identified through their reference lists. Finally, a hand-search of abstracts from the annual meetings of Digestive Disease Week, the American College of Gastroenterology, the European Crohn’s and Colitis Organization and the United European Gastroenterology Week, as well as of https://www.clinicaltrials.gov (accessed on 7 June 2022), was also employed up to 2022 to review the latest results of the ongoing clinical trials investigating this pharmacological mechanism. 

## 3. Molecular Pathway of S1P

S1P, a bioactive lipid mediator derived from mammalian cell membrane sphingolipids, is a pleiotropic molecule able to control several cellular processes by binding with varying affinities its five S1PR1-5, resulting in numerous and often complementary clinical effects [21]. However, S1P may act as an intracellular signaling molecule, and regulating functions have not been entirely clarified. In the last decade, the S1P/S1PR1 axis has sparked enormous interest for its property of controlling leukocyte trafficking [21]. Leukocyte trafficking and recruitment is a crucial component of the immune response, which starts with cellular tethering and rolling on the vessel endothelium, passes through integrin activation and adhesion, finally leading to the transmigration of leukocytes into the target tissue. In this contest, S1P plays a crucial role, working as a gatekeeper for the circulation of various immune cells, including B and T lymphocytes and natural killer and dendritic cells [17,21,22], based on a different concentration gradient between tissues and the circulatory system. The S1P gradient is created by the relatively low concentration of S1P in lymphoid organs in comparison with the lymph and tightly controlled by its synthesis and degradation mediated by two group of enzymes, sphingosine kinase (SphK1 and SphK2) and sphingoid base-specific phosphatases (SPPases), respectively [21]. S1P mostly binds to plasma carrier proteins, such as albumin and high-density lipoprotein (HDL), and albumin facilitates its transport, circulation and delivery in the target organs [17]. The loss of the S1P gradient causes the blockade of lymphocytes’ egress from the secondary lymphoid organs (i.e., lymph nodes) and the thymus [23,24]. S1PRs agonists induce the internalization of the receptor and its subsequent ubiquitination and proteasome-dependent degradation [23,24]. Among the sub-types of S1PRs, S1PR1 presents on T cells membranes, and its activation depends on the interaction with antigen-expressing dendritic cells [25,26]. Once the T cells have been activated in the lymphoid organs, S1PR1 expression decreases. S1PR1 activation proceeds in a STAT3-dependent manner and simultaneously triggers the RAS–ERK1/2 pathway and the PI3K-AKT and the PI3K–RAC pathways [22,25,26] (Figure 1). Several specific roles of the different S1PRs have been identified: S1PR1 is mainly involved in T cells regulation, S1PR2 regulates macrophage activation, S1PR3 endorses leukocyte rolling, S1PR4 controls the differentiation and activation of the dendritic cell (DC) and, finally, S1PR5 mainly participates in monocyte egressing from the bone marrow [15,27,28] (Figure 1). Once the lymphocytes are inside the lymphoid organs, the concentrations of cell-surface S1PR1 increase, so that the exit of lymphocytes from the lymphoid organs follows the S1P concentration gradient [17,21,22]. Conversely, S1PR1 expression is decreased after the activation of T cells [17,21,22].

Considering the role of S1PRs in the regulation of immune cells trafficking, activation and differentiation, their modulation has become an appealing novel target for the treatment of immune-mediated disorders.

The signal transduction of S1PRs occurs via the coupling of heterotrimeric G proteins, which, once activated, induce the RAS–ERK1/2 pathway, small GTPases, phospholipases, phosphatidylinositide 3-kinase (PI3K) and adenylyl cyclase. This cascade initiates multiple cellular effects (i.e., cellular migration and cytoskeletal rearrangement, cell proliferation and inhibition of apoptosis).

## 4. S1P Modulators in IBD: From Clinical Trials to Real World

### 4.1. Fingolimod

Fingolimod/FTY720 is an oral non-selective S1P modulator that binds S1PR1-3-4-5 [17]. Fingolimod was the first S1P modulator investigated in immune-mediated diseases [17] and was approved by the Food and Drug Administration (FDA) for the treatment of the relapsing multiple sclerosing [29]. In this setting, Fingolimod could reduce the circulating lymphocytes approximately by 70% [30,31].

In the field of IBDs, fingolimod exhibited encouraging results in animal models of IBDs. Indeed, treatment with this drug ameliorated chronic colitis in IL-10-gene-deficient mouse models and in Th1-mediated-2,4,6-trinitrobenzene-sulfonic-acid colitis [32,33]. Unfortunately, the cardiac safety profile is one of the main concerns regarding fingolimod, as transient cardiac events, such as bradycardia and atrioventricular block, have been detected [34,35]. In addition, during fingolimod treatment, a rising of liver enzymes was also observed, which resulted in drug discontinuation [17]. Therefore, due to the adverse events detected, fingolimod has not been tested in IBD human experimentations in favor of more selective S1PR modulators [17].

### 4.2. Ozanimod

Ozanimod/RPC1063 is a new oral S1P receptor modulator that binds with high selectivity to S1P1 and S1P5 receptors, preventing the mobilization of lymphocytes from peripheral lymphoid organs to inflammatory sites [36]. In regard to pharmacokinetics, ozanimod is characterized by slow absorption (median Tmax of 8 h; mean terminal elimination half-life of 21 h), and it is broadly metabolized into circulating active metabolites [17]. CYP2C8 has an important role in the drug’s metabolism, while CYP3A and P-glycoprotein contribute to a lesser extent [17].

The efficacy and safety data from two phase III trials had already led to the approval of ozanimod for the treatment of relapsing forms of multiple sclerosis [37,38].

Ozanimod lacks immunogenicity, has a high oral bioavailability, achieves peak concentrations after approximately 6 h and has a short half-life of about 19 h [39].

Regarding UC, the TOUCHSTONE phase II, double-blind, randomized clinical trial firstly reported that ozanimod achieved better clinical, endoscopic and histologic outcomes in patients with moderate to severe UC compared to placebo [40]. In particular, clinical remission at 8 weeks (the primary outcome) was achieved in 16% and 14% of the patients in the ozanimod 1 mg and 0.5 mg group vs. 6% in the placebo group (*p* = 0.048 and *p* = 0.14, respectively) [40]. Endoscopic healing (Mayo endoscopy sub-score ≤1) was observed in 34% and in 28% of the patients in the 1 mg and 0.5 mg ozanimod group, respectively, being 12% in the placebo group (*p* = 0.002 and *p* = 0.03, respectively) [40]. 

In the subsequent randomized, double-blind, placebo-controlled phase III study, the TRUE NORTH trial, in the treatment arm with ozanimod, higher clinical response (defined as a reduction in the total Mayo score of ≥3 points and ≥30% from baseline or in the three-component Mayo score of ≥2 points and ≥35% from baseline, as well as a reduction in the rectal-bleeding sub-score of ≥1 point or an absolute rectal-bleeding sub-score of ≤1 point) rates during both induction (47.8% vs. 25.9%, *p* < 0.001) and maintenance (60.0% vs. 41.0%, *p* < 0.001) were observed [41]. Clinical remission rates were significantly higher compared to placebo during both induction (18.4% vs. 6.0%, *p* < 0.001) and maintenance (37.0% vs. 18.5%, *p* < 0.001) [41]. Additionally, a significant improvement with ozanimod during both phases was found for other key secondary end points, such as maintenance of remission, mucosal healing, histologic remission and durable remission [41].

The long-term efficacy and safety of ozanimod in UC was recently reported in an interim analysis of the OLE of the TRUE NORTH trial: 34% of the whole study population and 55% of the responders continued to maintain a clinical response after 94 weeks of OLE [42].

As concerns CD, in the phase II multicenter, uncontrolled, prospective observer-blinded endpoint STEPSTONE trial, which enrolled 69 patients with moderately to severely active CD, treated with ozanimod 1 mg once daily for a 12-week-induction phase, a reduction from baseline in the Simple Endoscopic Score for Crohn’s Disease (SES-CD) was detected (mean change of −2.2); endoscopic remission (SES-CD score ≤4 and ≥2 point decrease in the SES-CD score with no sub-score >1) and endoscopic response (≥50% decrease in SES-CD) were achieved by 10.1% and 23.2% of the patients, respectively [43]. Moreover, the mean change in the Crohn’s Disease Activity Index (CDAI) score was of 130.4 [43]: 27 (39.1%) and 39 (56.5%) patients reached clinical remission (CDAI score <150) and clinical response (CDAI decrease of ≥100 from baseline), respectively [43]. Finally, the histological improvement in terms of the mean change from baseline in the Geboes Histology Activity Score and Robart’s Histopathology Index was also observed in the treated patients [43]. 

Several phase III placebo-controlled trials for ozanimod in patients with moderately to severely active CD (NCT03440385, NCT03464097, NCT03467958, NCT03440372, https://clinicaltrials.gov, accessed on 7 June 2022) are currently ongoing and/or actively recruiting.

With respect to adverse events (AEs), ozanimod demonstrated a good safety profile in IBD patients. Most of the AEs reported by II and III phase trials were not directly related to the drug, and the absolute number of the observed drug-related AEs was very low. 

In detail, in the TRUE NORTH trial, the serious AEs rate and the treatment discontinuation rate were higher in the placebo arm [41]. Although phase II and III trials data showed a favorable cardiac safety profile, a transient dose-dependent heart rate reduction was observed after the first dose of ozanimod (five patients with bradycardia during induction) [40,44]. To mitigate this risk, a seven-day gradual dose escalation upon treatment initiation is recommended [20,40,41,43]. Additionally, ozanimod is contraindicated in patients with cardiovascular and/or cerebrovascular disorders, in those with some conduction abnormalities (such as Mobitz type II second-degree or third-degree atrioventricular block, sick sinus syndrome or sinoatrial block, except those with a functioning pacemaker) or with severe untreated sleep apnea [20]. In case of other cardiac-related risk factors, ECG abnormalities or a drug history for medications which can cause bradycardia and a delay of cardiac conduction, a cardiologic evaluation is required [20].

In the STEPSTONE trial, no cases of bradycardia or arrhythmias occurred in the treated CD patients [43].

Due to a slightly increased risk of macular edema (two episodes during induction and one during maintenance), an ophthalmological evaluation at baseline is only recommended in patients at a high risk of macular edema (history of diabetes, uveitis) [40,41,43]. 

Furthermore, although a mean decrease of approximately 45% from baseline was observed in lymphocyte count (26.8% and 31.7% in the induction cohorts and 43.5% in the maintenance cohort), no severe infectious safety signals have been reported [40,41,43]. The infections’ rate in ozanimod-treated UC patients was comparable with placebo during induction (9.9% vs. 10.7%) and higher during maintenance (23% vs. 11.9%), but with serious infections occurring more frequently in the placebo arm (0.9% vs. 1.8%) [40,41]. Of note, the herpes zoster infections rate was higher with ozanimod during both induction (0.4% vs. 0%) and maintenance (2.2% vs. 0%) [40,41]. 

The elevation of transaminases occurred in 2.6% and in 1.6% of IBDs [43]-treated patients in the induction cohorts and in 4.8% during maintenance; however, it was described as usually transient (15–30 days) and unrelated to liver damage [40,41]. As regards drug interactions, the co-administration of MAO inhibitors and CYP2C8 inhibitors or inducers is contraindicated [20]. Additionally, ozanimod should not be used in subjects with hepatic impairment, end-stage renal disease and in pregnancy and breast-feeding due to lacking safety data in these groups [20]. Moreover, the coadministration of other immunomodulating drugs has not been investigated and, because of the additive risk of immunosuppression, should be avoided.

Based on these efficacy and safety data, ozanimod was approved by regulatory authorities for the treatment of moderate to severe UC. Newly published data on the OLE extension of the TRUE NORTH clinical trial have confirmed the efficacy and safety profile of ozanimod with rates of 0.9% of serious infections and of 0.4% of macular edema in the maintenance cohort [45]. Finally, two patients developed cancer during the OLE extension: one basal cell carcinoma and one rectal adenocarcinoma [45].

Recent data on long-term cardiac safety of ozanimod (from the phase 3 ulcerative colitis (UC) True North trial and multiple sclerosis (MS) 12-month SUNBEAM and 24-month RADIANCE trials) have shown that, in maintenance, cardiac-related AEs occurred in 1.3% (3/230) of treated patients in the continuous ozanimod group, and no clinically significant heart rate or electrocardiographic changes were related to chronic treatment up to month 24 of ozanimod therapy [46].

In the first real-world experience in relatively treatment-refractory UC patients, ozanimod showed a good tolerability and an efficiency and safety profile consistent with trials’ results [47]. Of note, despite the small sample size, efficiency was not significantly affected by prior therapy exposure [47].

The most recent post hoc analysis data on mucosal healing (MH) in UC patients treated with ozanimod showed that 44/230 (19.1%) patients achieved MH at week 10, being MH associated with ameliorated clinical outcomes (i.e., clinical remission, corticosteroid-free remission) at week 52 irrespective of previous treatment with anti-TNF [48]. Table 2 summarizes the most relevant efficacy and safety results of ozanimod in IBD.

### 4.3. Etrasimod

Etrasimod/APD334 (ArenaTM) is an investigational, once-daily orally administered selective S1P1R, S1P4R and S1P5R modulator. This molecule has been proven with promising results in several immune-mediated diseases (i.e., multiple sclerosis), including also IBDs. Concerning pharmacokinetics, etrasimod metabolization primarily involves hepatobiliary excretion thorough processes such as oxidation, dehydrogenation, glucuronidation and sulfation [17]. 

Preliminary data had reported that etrasimod rapidly decreases mean lymphocyte counts both in healthy volunteers and in UC patients with UC, with a subsequent lymphocyte recovery within 5% of baseline levels after 7 days from discontinuation [49,50]. 

As concerns UC, in the phase-2 OASIS study (NCT02447302, https://clinicaltrials.gov, accessed on 7 June 2022), including moderately-to-severely active UC with previous failure or intolerance to conventional or biologic therapy randomly assigned to etrasimod 1 mg (*n* = 52) or 2 mg (*n* = 50) or placebo (*n* = 54), etrasimod (2 mg/day) was shown to significantly improve clinical symptoms at week 12, assessed through a modified Mayo Clinic score (MCS) compared with a placebo (difference from placebo, 0.99 points; 90%CI 0.30–1.68; *p* = 0.009) [51]. In more detail, 33.0% of the UC patients receiving etrasimod (2 mg/day) compared with 8.1% of those in the placebo arm achieved clinical remission, defined as MCS ≤ 1 (*p* < 0.001) [51]. Among the secondary endpoints, a significantly higher amount of UC patients treated with etrasimod (2 mg/day) achieved endoscopic improvement at week 12 as compared with the placebo (41.8% vs. 17.8%; 90%CI 9.8%–39.0%; *p* = 0.003) [51].

Interestingly, histologic improvement (Geboes score <3.1) and histologic remission (Geboes score <2.0) were reported in 31.7% and in 19.5% of patients in the treatment arm (etrasimod 2 mg/day), respectively, vs. 10.2% and 6.1% in the placebo group, respectively (*p* = 0.006 and *p* = 0.03) [51]. With respect to safety, no life-threatening, drug-related adverse events (AE) and no deaths occurred during the study [51]. Cardiac AEs were reported in three cases receiving etrasimod 2 mg: a second-degree atrioventricular block type 1 (one patient) and first-degree atrioventricular block (two patients) [51].

Efficacy data were confirmed in the open-label extension (OLE) of the OASIS trial: at the end of treatment (week 46 or 52), 64% (72/112) and 33% (37/112) of the patients achieved clinical response and clinical remission, respectively, while 43% (48/112) of the patients showed endoscopic improvement, and endoscopic improvement was sustained from week 12 in more than two-thirds of patients [52]. Overall, steroid-free clinical remission was observed in 22% of patients at the end of treatment (week 46 or 52) [52]. In deeper detail, 65.2% (73/112) of the treated patients did not use oral corticosteroids at any time during the OLE [52]. 

The results of the OLE of the OASIS study reported that most AEs (94.4%) were of mild or moderate severity, and the most commonly occurring AEs were the worsening of UC (19%, 21/112 patients) and anemia (11%, 12/112 patients) [52]. Neither treatment-related serious infections nor infections of severity grade ≥3 were observed, and no patient died during the study [52]. Two cases of herpes zoster were reported [52]. 

Regarding cardiac AEs, one patient experienced heart rate lowering (with a nadir of 48 beats/min, grade 1 severity), without a need for dose change or treatment discontinuation [52]. Moreover, three patients receiving etrasimod 2 mg experienced a first-degree atrio-ventricular block, either clinically insignificant or of grade 1 severity; no patient discontinued etrasimod due to atrio-ventricular block [52].

The ELEVATE phase III trial (NCT03996369, NCT03945188, NCT03950232, https://clinicaltrials.gov, accessed on 7 June 2022) is currently enrolling moderately to severely active UC patients.

As concerns induction with etrasimod in UC patients, in the ELEVATE UC 12, clinical remission was observed in 24.8% of treated patients vs. 15.2% of the placebo group, respectively (*p* = 0.026) [53].

Preliminary results from the ELEVATE UC 52 have shown that the clinical remission rate was over three-times-higher in the treatment group with etrasimod compared with placebo at 12 weeks (27% vs. 7.4%; *p* ˂ 0.001) and over four-times-higher at 52 weeks (32.1% vs. 6.7%; *p* ˂ 0.001) [53]. Of note, all secondary efficacy endpoints, including endoscopic improvement, symptomatic remission, clinical response, and mucosal healing, were additionally met in both trials [53]. Furthermore, about 912 patients have been currently enrolled in the ongoing open-label extension study ELEVATE UC OLE, which is expected to be completed in 2027.

Regarding CD, CULTIVATE (NCT04173273, https://clinicaltrials.gov, accessed on 7 June 2022) is the ongoing phase II/III study aiming to enroll 1265 patients to evaluate the safety and efficacy of etrasimod, and it will assess, in its five sub-studies, the efficient dose of this molecule in the induction, maintenance/extension and long-term extension. Table 2 summarizes the most relevant efficacy and safety results of etrasimod in IBD.

### 4.4. Amiselimod

Amiselimod/MT-1303 is a novel oral selective S1P1 receptor modulator, which has been investigated for various immune-mediated diseases, such as multiple sclerosis, psoriasis, systemic lupus erythematosus and also IBD [54,55]. The higher affinity of Amiselimod for the S1P1 receptor than for S1P2-5 receptors makes this drug safer with respect to the cardiac profile compared to other non-selective S1P receptor modulators [54]. Indeed, amiselimod displayed a better cardiac safety than fingolimod in a phase I study [56].

Amiselimod and its active metabolite (S-amiselimod phosphate) have a half-life time 10-fold longer than ozanimod (380–420 h), reaching the steady-state concentration after about 10 weeks [17]. The pharmacokinetics profile of amiselimod appears more favorable for the cardiac profile, since low initial doses of the drug can lower the potential for bradycardia [57]. 

Regarding its efficacy in IBD, amiselimod was demonstrated to inhibit chronic colitis in mice models, preventing the egress of lymphocytes into the periphery and inhibiting the infiltration of Th1 and Th17 cells into the colon [58]. However, in a phase II, multicenter, randomized, placebo-controlled clinical trial that evaluated its efficacy and safety in moderate to severe CD, there was no significant difference in the proportion of subjects obtaining the primary endpoint (clinical response: CDAI 100 at week 12) between the amiselimod arm (0.4 mg daily) and the placebo arm (48% vs. 54.1%, respectively; OR 0.79; 95%CI 0.31–1.98) [59]. The elevated response in the placebo group and the smaller lymphocyte count reduction in comparison to multiple sclerosis trials are probably the principal reasons of the unmet endpoint in this study [59]. As concerns the safety profile of amiselimod in CD, during the study, no bradycardia or arrhythmic events were observed [59]. Both amiselimod and placebo arms displayed similar incidences of treatment-emergent AE and serious adverse reactions, while the amiselimod group exhibited a higher incidence of serious adverse events (amiselimod 15.4% vs. placebo 2.6%) [59]. The development of amiselimod was discontinued by Biogen, and its rights were returned to Mitsubishi Tanabe Pharma [54]. Subsequently, the Salix Pharmaceuticals signed an exclusive licensing agreement with Mitsubishi Tanabe Pharma to develop and commercialize amiselimod in IBD [60], and a phase 2 RCT evaluating its efficacy and safety in mid to moderate ulcerative colitis is ongoing (NCT04857112, https://clinicaltrials.gov, accessed on 7 June 2022).

## 5. Discussion

This review elucidated the role of S1P modulation as the emerging therapies’ target in the treatment of IBD. S1P1 agonists are a new generation of oral small molecules, with a short half-life and no risk of immunogenicity.

As emerges from the above exposed data, S1PRs modulators have been demonstrated to be well tolerated [40,41,43,51,52]. The regulation of S1P2R and S1P3R may be associated with cardiovascular, pulmonary, and theoretical cancer-related risks. Nevertheless, regarding cardiac AE, the available data indicate that there is no need for dose titration at the beginning of treatment because of any arrythmia concern (i.e., bradycardia, conduction abnormalities). Regarding the infectious risk, while the need of tuberculosis and hepatitis B screening is currently debated, the varicella zoster virus (VZV) antibody titer is recommended, and, if negative, the completion of the vaccination series is advised before starting the therapy with S1P modulators.

However, the long-term safety profile of these agents requires further monitoring to establish possible dose dependency, defining possible inter-individual variability in terms of pharmaco-kinetics and reversibility.

Encouraging preclinical and clinical data exploring the efficacy of S1P modulators have demonstrated the inhibition of lymphocyte egress from the lymph nodes as a valid pharmacological strategy to induce remission in IBD. Rapid response [61], remission [61] and sustained efficacy have been proven both in terms of clinical remission and endoscopic improvement in both UC and CD patients [40,41,43,51,52]. 

S1P modulators are not limited by potential immunogenicity, in contrast to most of the currently approved biologic agents.

An additional advantageous aspect of S1P1 modulators is the rather rapidity for lymphocytes to return to normal levels after drug withdrawal [49,50], which becomes relevant for the daily management of patients, in case of therapy swap or mandated interruption for scheduled and/or emergency surgeries.

Further benefits of this drugs’ class include the maintenance of T cell effector’s memory response and possible low manufacturing cost.

Moreover, several studies have demonstrated that circulating S1P activate endothelial S1PRs, regulating angiogenesis and stabilizing blood vessels in the development and homeostasis [62,63]. Finally, in vivo and in vitro experiments have shown that S1P treatment significantly increased levels of the E-cadherin protein and mRNA in intestinal epithelial cells, improving the barrier integrity [64]. These complementary effects might reveal the determinant for the long-term efficacy and overall preservation of the gut functions of this drugs’ class. 

In our view, the open issues regarding S1P modulators mainly concerns their positioning within current treatment algorithms, depending on disease location (ileal, ileocolonic or purely colonic involvement) and their potential as combination therapies for the treatment of moderate-to-severe, multi-failure IBD patients [65]. The benefit of S1PR modulators in IBD patients with uncontrolled extra-intestinal manifestations needs further investigation, as well as their potential role in the treatment of operated patients or pediatric-onset IBDs. Data from the post-hoc analysis from the pivotal phase 3 TRUE NORTH trial suggest that the most effective place in therapy of S1P modulators might be in biologically naïve patients [66]. 

Recent research comparing the efficacy and safety data of ozanimod and ustekinumab in UC patients reported a comparable response in both biologically naïve and biologically exposed patients, with no statistically significant differences between the two drugs [67]. However, the ozanimod groups showed lower rates of endoscopic improvement (OR and 95%CI, 0.26 and 0.10–0.72, respectively) compared to ustekinumab [67]. Moreover, ozanimod displayed significantly lower rates of infectious AEs compared to ustekinumab (RD and 95%CI, −24.9% and −34.6 to −15.2%, respectively) [67].

The addressing of these unanswered questions warrants head-to-head trials and comparative effectiveness studies. Finally, predictive biomarkers for patients’ molecular profiling, to predict potential responders as well as to monitor responsiveness, are highly needed to facilitate and better support therapeutic strategies.

## Figures and Tables

**Figure 1 biomedicines-10-01735-f001:**
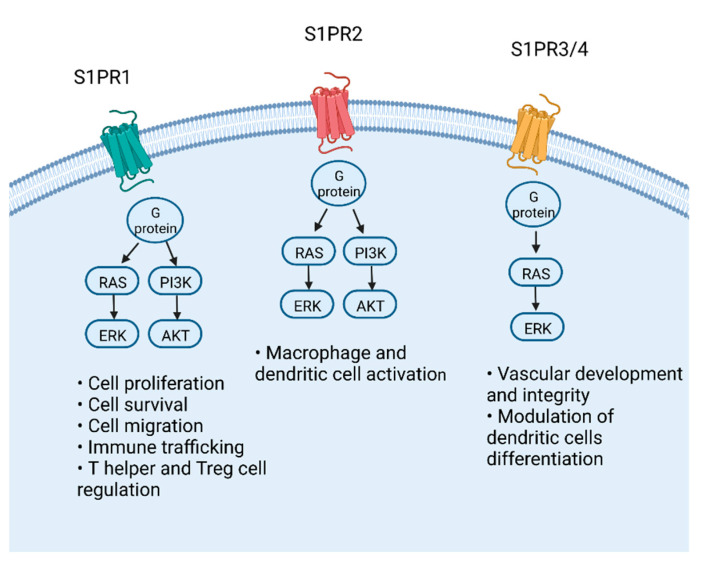
Cellular processes and the molecular pathway of S1P receptors.

**Table 1 biomedicines-10-01735-t001:** Principal characteristics and development phase of S1P receptor modulators in IBD.

Molecule	Pharmacological Mechanism	Administration	Development
Fingolimod	non-selective, S1PR1-3-4-5	Oral	Never tested in humans
Ozanimod	S1PR1, S1PR4 and S1PR5	Oral	UC: FDA and EMA approved CD: Phase II/III recruiting
Etrasimod	S1PR1 and S1PR5	Oral	UC: Phase III completed CD: Phase III recruiting
Amiselimod (MT-1303)	S1PR1	Oral	CD: Phase II completed

IBD: inflammatory bowel diseases; UC: ulcerative colitis; CD: Crohn’s disease; S1P: sphingosine 1-phosphate; S1PR: sphingosine 1-phosphate receptor.

**Table 2 biomedicines-10-01735-t002:** Efficacy and safety of ozanimod and etrasimod in IBD.

Molecule	Efficacy in CD	Efficacy in UC	Safety in CD	Safety in UC
Ozanimod	39.1% and 56.5% of the patients reached clinical remission (CDAI score <150) and clinical response (CDAI decrease of ≥100 from baseline), respectively	Clinical remission at 8 weeks achieved in 16% and 14% of the patients in the 1 mg and 0.5 mg group Endoscopic healing * observed in 34% and in 28% of the patients in the 1 mg and 0.5 mg group. Clinical remission rates were significantly higher compared to placebo during induction (18.4% vs. 6.0%, *p* < 0.001) and maintenance (37.0% vs. 18.5%, *p* < 0.001). At 94 weeks 34% of the whole study population and 55% of the responders continued to maintain clinical response.	In the STEPSTONE trial, no cases of bradycardia/arrhythmia occurred in the treated CD patients	Transient dose-dependent heart rate reduction at induction. Infections’ rate in ozanimod-treated UC patients was comparable with placebo during induction (9.9% vs. 10.7%) and higher during maintenance (23% vs. 11.9%).
Etrasimod	ongoing phase II/III study	Clinical remission assessed by 33.0% after 12 weeks of treatment (2 mg/day). Endoscopic improvement at week 12 was significantly higher as compared with placebo (41.8% vs. 17.8%; *p*= 0.003). Steroid-free clinical remission observed in 22% of patients at end of treatment (week 46/52)	ongoing phase II/III study	One patient with a second-degree atrioventricular block type 1 and two patients with first-degree atrioventricular block. Neither treatment-related serious infections nor infections of severity grade ≥3 were observed. Two cases of herpes zoster were reported

* Mayo endoscopy sub-score ≤1, IBD: inflammatory bowel diseases; UC: ulcerative colitis; CD: Crohn’s disease.

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
