# Peer review of "Sphingosine 1-Phosphate Modulation in Inflammatory Bowel Diseases: Keeping Lymphocytes Out of the Intestine"

_biomedicines, 2022, doi:10.3390/biomedicines10071735_

Round 1
Reviewer 1 Report
The topic undertaken by the authors is very interesting and highly desirable since there is no cure for IBDs. However, I feel the manuscript lacks some significant information to fully comprehend the topic:
Introduction:
The table should be moved to the part where it is cited.
Molecular pathway of S1P:
Line 102 – What about the concentration of S1P in plasma? And the regulation of the S1P receptors?
What about the action of S1P on the maintenance of vascular barrier integrity and its modulation of adherens junctions – is it also affected by S1P receptor modulators and how can it affect the leucocytes flow. – are there any studies? – or maybe discuss that in the Discussion section.
S1P modulators in IBD: from clinical trials to the real world:
More information about pharmacodynamics and pharmacokinetics of described drugs – if are available and their metabolism.
I don't feel qualified to judge about the English language and style but in my opinion, the paper has to be read again and spelling/missing words/unneeded words should be corrected.
Author Response
Response to the editor(s) of ‘Biomedicines’ Journal
Dear Editor(s),
We sincerely thank you for giving us the opportunity to consider for re-submission and potential publication a revised version of our original manuscript entitled “Sphingosine 1-phosphate modulation in inflammatory bowel diseases: keeping lymphocytes out of the intestine" by Dal Buono et al.
We kindly thank the Reviewers for the precious comments. We are pleased to know that you appreciated the topic of our manuscript. The manuscript has been significantly revised and improved according to the received suggestions. Included below you can find a point-by-point response to the remarks.
Sincerely,
Alessandro Armuzzi, Professor
alessandro.armuzzi@hunimed.eu
Reviewer#1
The topic undertaken by the authors is very interesting and highly desirable since there is no cure for IBDs. However, I feel the manuscript lacks some significant information to fully comprehend the topic:
Re: Thank You for Your comment. We are glad to know that You appreciated our paper, below a point-by-point response to the remarks.
Introduction:
The table should be moved to the part where it is cited.
Re: Thanks for Your comment. We moved the Table as suggested.
Molecular pathway of S1P:
Line 102 – What about the concentration of S1P in plasma? And the regulation of the S1P receptors?
Re: Thanks for Your comment. We added details on both topics as suggested.
What about the action of S1P on the maintenance of vascular barrier integrity and its modulation of adherens junctions – is it also affected by S1P receptor modulators and how can it affect the leucocytes flow. – are there any studies? – or maybe discuss that in the Discussion section.
Re: Thanks for Your comment. We mentioned and commented on that in the discussion section as suggested.
S1P modulators in IBD: from clinical trials to the real world:
More information about pharmacodynamics and pharmacokinetics of described drugs – if are available and their metabolism.
Re: Thanks for Your comment. We have added details on pharmacodynamics and pharmacokinetics within the drugs’ paragraphs accordingly.
I don't feel qualified to judge about the English language and style but in my opinion, the paper has to be read again and spelling/missing words/unneeded words should be corrected.
Re: Thanks for Your comment. We had a mother tongue college read the paper for revising the English language.
Reviewer 2 Report
This review is well written, up-to-date and with attention to details, presenting and analyzing aspects regarding Sphingosine 1-phosphate receptor modulators (especially efficacy and safety), in the treatment of inflammatory bowel disease. Since the manuscript is wonderful, I only have some minor suggestions/comments:
1. Abstract: a. The authors wrote: “Treatment purpose is to reduce symptoms, progression of the disease and improve quality of life.” I suggest here to revise the purposes, according to the "STRIDE II" targets. b. The authors wrote: “All S1P modulators are reported to be safe and effective in the treatment of both UC and CD”. However, there are no studies with Fingolimod in humans with IBD (due to reported adverse events). Please revise. c. The authors wrote: “and from ongoing clinical trials on the safety and efficacy of selectivity of S1P modulators in the treatment of IBD”. I highly appreciate including ongoing trials in the manuscript; however, “Methods” in the manuscript do not include this aspect (please see below).
2. Methods: Just the search on PubMed and EMBASE was written. There is nothing about searching data in ongoing clinical trials - https://www.clinicaltrials.gov. Please add.
3. Molecular pathway of S1P – Adding a figure with these mechanisms would be very useful.
4. Ozanimod and Etrasimod – a table summarizing data about their efficacy and safety in both UC and CD (Ozanimod) and UC (Etrasimod) should be added.
5. Paragraph “Discussion has to be numbered as “5”, not “4”.
Author Response
Response to the editor(s) of ‘Biomedicines’ Journal
Dear Editor(s),
We sincerely thank you for giving us the opportunity to consider for re-submission and potential publication a revised version of our original manuscript entitled “Sphingosine 1-phosphate modulation in inflammatory bowel diseases: keeping lymphocytes out of the intestine" by Dal Buono et al.
We kindly thank the Reviewers for the precious comments. We are pleased to know that you appreciated the topic of our manuscript. The manuscript has been significantly revised and improved according to the received suggestions. Included below you can find a point-by-point response to the remarks.
Sincerely,
Alessandro Armuzzi, Professor
alessandro.armuzzi@hunimed.eu
Reviwer#2
This review is well written, up-to-date and with attention to details, presenting and analyzing aspects regarding Sphingosine 1-phosphate receptor modulators (especially efficacy and safety), in the treatment of inflammatory bowel disease. Since the manuscript is wonderful, I only have some minor suggestions/comments:
Re: Thank You for Your comment. We are glad to know that You appreciated our paper, below a point-by-point response to the remarks.
- Abstract:
- The authors wrote: “Treatment purpose is to reduce symptoms, progression of the disease and improve quality of life.” I suggest here to revise the purposes, according to the "STRIDE II" targets.
Re: Thanks for Your comment. We corrected accordinlgy as suggested.
- The authors wrote: “All S1P modulators are reported to be safe and effective in the treatment of both UC and CD”. However, there are no studies with Fingolimod in humans with IBD (due to reported adverse events). Please revise.
Re: Thanks for Your comment. We corrected accordingly as suggested.
- The authors wrote: “and from ongoing clinical trials on the safety and efficacy of selectivity of S1P modulators in the treatment of IBD”. I highly appreciate including ongoing trials in the manuscript; however, “Methods” in the manuscript do not include this aspect (please see below).
Re: Thanks for Your comment. We improved the methods section as suggested.
- Methods: Just the search on PubMed and EMBASE was written. There is nothing about searching data in ongoing clinical trials - https://www.clinicaltrials.gov. Please add.
Re: Thanks for Your comment. We improved the methods section as suggested
- Molecular pathway of S1P – Adding a figure with these mechanisms would be very useful.
Re: Thanks for Your comment.
- Ozanimod and Etrasimod – a table summarizing data about their efficacy and safety in both UC and CD (Ozanimod) and UC (Etrasimod) should be added.
Re: Thanks for Your comment. We have added a dedicated table (Table 2) as suggested.
- Paragraph “Discussion has to be numbered as “5”, not “4”
Re: Thanks for Your comment. We have corrected accordingly.